# Concurrent Infection with SARS-CoV-2 and *Pneumocystis jirovecii* in Immunocompromised and Immunocompetent Individuals

**DOI:** 10.3390/jof8060585

**Published:** 2022-05-30

**Authors:** Francesca Gioia, Hanan Albasata, Seyed M. Hosseini-Moghaddam

**Affiliations:** Transplant Infectious Diseases Program, Ajmera Transplant Centre, Toronto General Hospital, University Health Network, University of Toronto, Toronto, ON M5G 2N2, Canada; francesca.gioia@uhn.ca (F.G.); hanan.albasata@uhn.ca (H.A.)

**Keywords:** severe acute respiratory syndrome coronavirus 2, *Pneumocystis jirovecii* pneumonia, corticosteroids, lymphopenia

## Abstract

Coronavirus disease 2019 (COVID-19) may occur with concurrent infections caused by bacterial and fungal microorganisms. This systematic review evaluated studies reporting concomitant COVID-19 and *Pneumocystis jirovecii* pneumonia (PJP). We found 39 patients (74% male, median age: 56.8 (range: 11–83) years), including 66% immunosuppressed individuals (23% HIV-infected and 41% on long-term corticosteroid therapy). Patients were characteristically severely ill (mechanical ventilation: 70%), associated with 41% mortality. The median lymphocyte count was 527 cells/mm^3^ (range: 110–2200), and the median CD4+ T cell count was 206 cells/mm^3^ (range: 8–1021). We identified three patterns of concurrent COVID-19 and *P. jirovecii* infection. The first pattern (airway colonization with a low burden of *P. jirovecii*) does not seem to modify the COVID-19 course of illness. However, *P. jirovecii* superinfection, typically occurring weeks after COVID-19 diagnosis as a biphasic illness, and *P. jirovecii* coinfection characteristically results in progressive multilobar pneumonia, which is associated with poor outcomes. To support this categorization, we reported three patients with concurrent PJP and COVID-19 identified in our institution, presenting these clinical scenarios. The diagnosis of PJP requires a high index of suspicion, since clinical and radiological characteristics overlap with COVID-19. Observational studies are necessary to determine the PJP burden in patients with COVID-19 requiring hospitalization.

## 1. Introduction

*P. jirovecii* is a common microorganism that can be detected from respiratory specimens of asymptomatic healthy adults and children [1,2]. The prevalence of *P. jirovecii* in the general population is not well established. Several studies in different regions of the world have reported a colonization rate from 0% to 65%, depending on the sample and the laboratory technique used for the detection of *P. jirovecii* and also on the population studied [3,4,5]. Data from Chile showed *P.jirovecii* in 61% of autopsy specimens in healthy adults who died in the community of violent causes [6].

Asymptomatic acquisition of *P. jirovecii* probably occurs in childhood. Various studies showed the presence of antibodies against *P. jirovecii* in children [7,8]. Vargas et al. described seroconversion in 85% of healthy children aged between 2 and 20 months, and a *P. jirovecii* DNA was identified in nasopharyngeal aspirates of 32% of infants until 2 years of age during episodes of mild respiratory infection [9]. According to a study conducted in Spain, *P. jirovecii* seroprevalence increased with age, reaching 52% at age 6, 66% at age 10, and 80% at age 13 [10]. Hence, the prevalence of infection with *P. jirovecii* appears to increase with age.

In immunocompromised hosts, *P. jirovecii* can cause *Pneumocystis jirovecii* pneumonia. A growing number of studies have shown evidence of PJP in HIV-negative immunocompromised patients [11]. The most recognized risk factors for developing PJP in non-HIV patients included immunosuppressive therapy, such as glucocorticoid use, and CD4+ lymphocytopenia [12].

Severe acute respiratory syndrome coronavirus 2 (SARS-CoV-2) infection may cause a progressive coronavirus disease 2019 (COVID-19). A wide spectrum of the immune response to SARS-CoV-2 is expected, ranging from a severe systemic inflammation to a marked systemic immune suppression (i.e., immunoparalysis) [13]. Recent studies showed that profound depletion of CD4+ lymphocytes and NK cells is common in patients with severe COVID-19 [14]. Immunosuppressive therapy and treatment with glucocorticoids that are routinely considered for the management of severe COIVD-19 cause further immune suppression and have the potential to promote the development of opportunistic infection.

From the beginning of the pandemic, several studies have shown that severely ill COVID-19 patients are predisposed to concurrent infections caused by different microorganisms [15,16]. COVID-19 likely increases the risk of invasive fungal infection (IFI). Treatment of patients with severe COVID-19 using systemic glucocorticoids and immunosuppressive agents [17,18] can increase the risk of IFIs [19,20,21,22]. Glucocorticoid therapy could be continued for 10 days or until discharge. In this population, the occurrence of IFIs also may be the result of a variety of factors, including epithelial barrier damage, admission to the intensive care unit (ICU), invasive mechanical ventilation (IMV), and prolonged hospitalization [23].

Some IFIs, such as invasive aspergillosis, systemic candidiasis, and mucormycosis, have been frequently reported in patients with severe COVID-19 [24,25,26,27,28]. Several patients with moderate-to-severe COVID-19 have been reported with *P. jirovecii* simultaneous infection, in which lymphopenia and corticosteroids appear to be the main PJP risk factors [29].

Furthermore, *P. jirovecii* simultaneous infection in COVID-19 patients represents a significant diagnostic challenge for several reasons [12]. The clinical presentation and radiological futures of the two diseases overlap [30], making it difficult to distinguish between them, especially in high-risk patients receiving glucocorticoid therapy [31]. In COVID-19 severely ill patients, the microbiological diagnosis of PJP could be a problem. The diagnosis of SARS-CoV-2 infection is routinely made on upper respiratory tract specimens, while the diagnosis of PJP requires sampling from lower airways. Thus, concomitant or superimposed PJP are likely underestimated in COVID-19 patients. The diagnostic tools for PJP and other IFIs are not well validated in upper respiratory specimens [32,33]. In addition, the current clinical diagnostic algorithm for PJP does not apply to COVID-19 patients, making it difficult to establish if the detection of *P. jirovecii* should be interpreted as real infection or colonization [29,30,31,32,33,34,35].

This narrative review aims to analyze the information available in the literature about COVID-19 and simultaneous infection with *P. jirovecii,* describing the diagnostic challenges, clinical characteristics, outcomes, and prognosis.

## 2. Materials and Methods

We performed a qualitative and comprehensive review of the literature using a combination of database-specific subject headings and text words for the main concepts of COVID-19 and *P. jirovecii* concomitant infection. The strategy was initially designed for Pubmed and then translated for Embase. We included studies published in the databases from 1 January 2022 to 19 April 2022.

We included case reports, case series, and observational studies, including hospitalized COVID-19 patients with *P. jirovecii* concomitant infection. Conference proceedings and books were removed. English, Italian, or Spanish studies were included. See Appendix A for full search strategies.

We only included studies with a confirmed diagnosis of COVID-19 using real-time polymerase chain reaction (RT-PCR) from a nasopharyngeal or oropharyngeal swab or BAL. The diagnosis of PJP was considered ‘proven’ if *P. jirovecii* was detected on BAL with immunofluorescence microscopy (IFM); ‘probable’ in the presence of host factors, clinical features, and microbiological evidence (PCR for *P. jirovecii* on respiratory samples or serum Beta-D-Glucan assay (BDG)); and ‘possible’ in the absence of microbiological evidence, according to the European Organization for Research and Treatment of Cancer and the Mycoses Study Group (EORTC/MSGERC) criteria [12].

Detailed information about the studies, demographics, and clinical characteristics were collected and are summarized in Table 1. The collected data included age, comorbidities, radiological findings, and absolute lymphocyte count during PJP diagnosis; lowest CD4+ T cell count; long-term corticosteroid exposure before admission; and other immunosuppressive medications or conditions.

### Clinical Patterns of PJP

Considering the course of illness, clinical data, and radiographic findings, we defined coinfection as microbiologic evidence of infection with *P. jirovecii* during moderate-to-severe COVID-19 requiring anti-PJP therapy. Patients with COVID-19 who improved following treatment (e.g., glucocorticoid therapy) and subsequently developed microbiologically proven PJP (i.e., biphasic illness) were defined as a superinfection. We categorized patients with a low burden of *P. jirovecii* (e.g., <10^3^ in quantitative PCR), which did not require treatment, as airway colonization. Using these definitions, we subsequently presented three patients with COVID-19 and a laboratory diagnosis of PJP.

## 3. Results

### 3.1. Literature Review

#### Demographic and Clinical Characteristic of Patients

Twenty case reports and two observational studies described 39 patients with concurrent *P. jirovecii* and SARS-CoV-2 infection (74% male, median (range) age: 56.8 (11–83)). A total of 66.6% (26/39) of patients were immunosuppressed at the baseline (e.g., HIV-infection, receiving immunosuppressive drugs). Among immunocompromised patients, 23% (9/39) were HIV-infected. Forty-one percent (16/39) of patients received long-term corticosteroid therapy before being diagnosed with COVID-19. Other immunosuppressive therapies included leflunomide, tacrolimus, mycophenolate mofetil, fludarabine, cyclophosphamide, and rituximab.

### 3.2. Diagnosis and Outcome

PJP was diagnosed in 41% (16/39) of cases based on BAL specimens, in 7.6% (3/39) on tracheal aspirate specimens (TA), and 12.8% (5/39) on sputum samples. In two cases, histopathological or cytological findings were described. Among the 10 cases reported by Alaino et al. [34], no source of the low respiratory tract (LRT) sample was specified. A possible PJP diagnosis was established in four patients with no microbiological evidence.

PCR was the most common method for PJP diagnosis (66.6%), followed by microscopy staining and IFM (23%). Data for BDG were available for 17 patients (median: 260 ng/L). *Aspergillus* spp. coinfection was detected in 15.3% (6/39) patients.

All patients had radiographic findings compatible with PJP. The most common radiologic feature was ground-glass opacity (GGO), followed by lobar infiltration and pleural effusions.

The median lymphocyte count during PJP diagnosis was 527 cells/mm^3^ (range: 110–2200), and the median CD4+ T cell count was 206 cells/mm^3^ (range: 8–1021). Most patients were critically ill, and 70% (26/39) needed mechanical ventilation. PJP was treated in 76.9% of patients (30/39), most with sulfamethoxazole-trimethoprim (TMP-SMX) (28/39). In 2/39 cases, echinocandins were used. The mortality rate was 41% (16/39).

### 3.3. Brief Description of Three Cases in Our Institution

In our institution, we identified three cases of COVID-19 and simultaneous PJP diagnosis.

### 3.4. CASE 1 and 2: Superinfection with P. jirovecii

The first case was a 63-year-old male with a past medical history of chronic lymphocytic leukemia diagnosed in 2016 and who received treatment with obinutuzumab and venetoclax. He also had hypertension and ischemic heart disease. He had no prior history of steroid use. He received two doses of the COVID-19 vaccines in April and May 2021. He initially presented with a moderate cough associated with shortness of breath at the beginning of July 2021. He underwent BAL, and his SARS-CoV-2 test was reported positive. He did not receive any specific treatment for COVID-19. He was subsequently readmitted to our hospital at the end of July 2021 due to shortness of breath, anosmia, and ageusia. He was afebrile and normotensive. His oxygen saturation was 93% using 2 L/min O2 via a nasal cannula. The initial laboratory evaluation revealed leukocytosis and neutrophilia. The chest CT showed a worsening of ground glass opacity (GGO) in both upper lobes, with a reverse halo appearance, consolidation, and bibasal pleural effusion. He underwent the second BAL, and his *P. jirovecii* PCR test was positive (1123 copies/mL). SARS-CoV-2 was still detectable in the BAL-PCR test. He was treated with trimethoprim-sulfamethoxazole (TMP-SMX) associated with tapering doses of prednisone for 21 days and subsequently discharged home.

The second patient was a 43-year-old male with primary refractory acute myeloid leukemia, treated with intravenous cytarabine and oral hydroxyurea. He was also on 20 mg of dexamethasone from October 2021, receiving a tapering dose for more than four weeks. He did not receive SARS-CoV-2 vaccines. In December 2021, he developed mild respiratory symptoms, and his nasopharyngeal swab of SARS-CoV-2 was positive. At this time, he did not receive any specific treatment for COVID-19. He was admitted with fever (T: 38.8 °C), dyspnea, and chest pain one month later. His oxygen saturation was 95% while receiving 2 L/min O2 via a nasal cannula. His nasopharyngeal swab of SARS-CoV-2 was still positive. His laboratory evaluation revealed severe neutropenia and lymphopenia. The chest CT showed multilobular consolidation, pulmonary nodules, and GGO. Despite receiving glucocorticoid therapy and remdesivir for COVID-19 pneumonia, his symptoms and oxygen requirement did not improve. Three days after the admission, he underwent a bronchoscopy, which confirmed the diagnosis of PJP (PCR for *P. jirovecii*: 10,700 copies/mL). In addition, BAL was positive for COVID-19 (Omicron variant). Recovery began after treatment with TMP-SMX, and the patient was discharged home.

### 3.5. CASES 3-Coinfection with P. jirovecii

The third case was a 62-year-old male admitted in December 2020 for induction of acute lymphoblastic leukemia with blinatumomab and ponatinib.

He had a past medical history of hypertension and hepatitis B treated with entecavir. He had a prior history of steroid use (≥30 mg of prednisone daily) for more than 4 weeks. He did not receive a COVID-19 vaccination.

He presented with febrile neutropenia and hypoxia, requiring oxygen in a nasal cannula of 2 L/min. Laboratory evaluation revealed severe neutropenia and a normal lymphocyte count. His chest CT showed extensive consolidation in the right lower lobe. He underwent a bronchoscopy, and both SARS-CoV-2 PCR and *P. jirovecii* PCR (3250 copies/mL) were positive. The patient received a 21-day course of treatment with TMP-SMX associated with adjunctive glucocorticoid therapy. He was discharged home one month later.

## 4. Discussion

The present review showed that PJP could be at least a component of the causal complex of pneumonia in patients with severe COVID-19. Treatment regimens for the management of severe COVID-19 may cause an immunodeficiency state. Additionally, SARS-CoV-2 infection may cause dysregulation of immune system function. We identified 39 COVID-19 patients in the literature with simultaneous infection with *P. jirovecii*. Almost half of them had immunosuppressive conditions, including HIV-infection and long-term corticosteroid therapy, before PJP diagnosis. Lymphocytopenia was a common finding in almost all patients, and they frequently had CD4+ T cell counts less than 400 cells/mm^3^. The most common radiological feature described was GGO and diffuse bilateral infiltration. PJP was diagnosed by a specimen source of BAL primarily, and PCR was used in more than half of the LRT specimens. Most patients received anti-PJP therapy, predominantly TMP-SMX; however, the patients were severely ill, wih most requiring IMV, and the mortality rate was high. The overall mortality in reported cases was considerably high, indicating that PJP is associated with an unfavorable outcome in patients with COVID-19, and a timely diagnosis may be lifesaving.

Due to similarities in radiological and clinical findings among patients with COVID-19 and PJP, the differential diagnosis is challenging. COVID-19 and PJP may present overlapping clinical and radiological manifestations [31,55,56]. Patients present with similar features of dry cough, exertional dyspnea associated with O2 desaturation, and relatively normal chest auscultation [37,57]. The chest CT findings for both conditions are similar, with a GGO pattern and thickening of the interlobular septum [58,59]. The presence of pleural effusions, lung cysts, and pneumothorax should support a PJP diagnosis [60,61,62]. The gold standard for PJP diagnosis is based on microscopic visualization of *P. jirovecii* in LRT, especially BAL, due to its higher sensitivity [32,33]. On the other hand, COVID-19 diagnosis does not need bronchoscopy. Moreover, BAL is usually avoided in patients with COVID-19 due to the risk of O2 desaturation and SARS-CoV-2 aerosolization. All these factors undoubtedly cause underestimation of PJP incidence in COVID-19 patients. Therefore, it may be reasonable to include PJP in the differential diagnosis of patients with severe COVID-19 and a lack of improvement following glucocorticoid therapy.

Lymphocytopenia and a low CD4+ T cell count are common futures in patients with severe COVID-19, which correlate with the severity of illness [63,64]. Patients with a poor CD4+ T cell-count are susceptible to *P. jirovecii* colonization and PJP [12,65,66,67]. Many studies suggested that the immune dysregulation in patients with severe COVID-19, especially CD4+ lymphopenia, may increase the risk of PJP [30,35,68,69]. Patients with PJP and SARS-CoV-2 infection may have lymphopenia without any other classical PJP risk factors [35]. Some studies have cast doubt on the relationship between the lymphocytopenia encountered in severe COVID-19 and PJP, since lymphopenia in these patients may be transient. Blaize et al. performed 423 *P. jirovecii* PCR assays on respiratory samples obtained from 145 patients with severe SARS-CoV-2 infections; most of them (79%; 113/143) had a mild lymphocytopenia (1000 cells/mL), and almost all (99.3%; 420/423) had a negative *P. jirovecii* PCR test [54] However, this level of lymphopenia is not generally considered a PJP risk factor, and it would be of interest to have a comparable analysis using a more severe lymphopenia (<800 cells/mL). Further investigation is necessary to determine the risk of PJP in different levels of lymphopenia.

In addition to lymphopenia, COVID-19 also produces epithelial damage with resulting impaired mucociliary clearance, which may increase the susceptibility to PJP, as seen in other IFIs [26,70]. Recently, a study by Kottom et al. demonstrated the role of airway epithelial cells in binding and subsequently recognizing microorganisms, such as *P. jirovecii,* and in the initiation of the immune response [71].

Simultaneous infection with *P. jirovecii* has been shown with other respiratory viral pathogens. There is evidence that *P. jirovecii* is responsible for 7% of simultaneous infections among non-immunocompromised patients admitted to ICUs with influenza [72]. Single-center studies showed tha *P. jirovecii* might be detected in up to 9% of respiratory samples from patients with COVID-19 admitted to ICUs [34,54,73]. The noteworthy point is that there was no systematic approach to detect concomitant PJP in all those studies. The data were collected at the beginning of the pandemic when steroids and IL-6 inhibitor biologics were not widely used in patients with COVID-19. Currently, patients with severe COVID-19 routinely receive glucocorticoid therapy for 10 days or until hospital discharge, and a higher rate of infection with *P. jirovecii* is expected.

The present review identified almost three patterns of simultaneous infection with *P. jirovecii* (Table 2). The first pattern is more likely to be airway colonization with *P. jirovecii.* In the study of Alanio et al., 9 of 108 patients (9.3%) had a positive *P. jirovecii* PCR test, and more than a half (n = 6) did not receive any treatment [34]. Similarly, Blaize et al. described two patients with severe COVID-19 and a positive *P. jirovecii* PCR test from BAL specimens who improved without PJP treatment. Both patients had a low burden of microorganisms (<3 log). The serum BDG level was tested in one patient and was negative. None of these two patients presented with classical PJP host factors [54].

The second pattern of PJP is superinfection, occurring days or weeks after COVID-19 symptom onset as a biphasic illness. This presentation is well described by Gentile et al. [42]. All patients in this cohort developed PJP after clinical resolution of COVID-19 pneumonia, and the median time between COVID-19 symptoms and PJP diagnosis was 40 days. There were no pre-existing PJP risk factors in any of those patients except one, but all received at least 2 weeks of high-dose steroids, and the mean CD4+ T cell count at PJP diagnosis was 141 cell/mm^3^ (range: 77–953.5). Similarly, Gerber et al. [40] described four cases; two of them did not have any risk factor for PJP before the COVID-19 diagnosis, and developed PJP 25 and 40 days after admission for COVD-19. Both patients received glucocorticoid therapy for COVID-19 (dose and duration not specified). Cases 1 and 2 in our institution also seemed to have the same pattern of PJP superinfection, even if case 2 had a previous history of glucocorticoid therapy not related to COVID-19.

The third pattern of presentation is *P. jirovecii* coinfection, resulting in progressive multilobar pneumonia. In most of these patients, long-term exposure to immunosuppressants or a severe immunodeficiency were present as PJP risk factors. In the present review, more than half of patients were immunocompromised at the baseline. These patients frequently presented with worsening of respiratory illness despite the treatment for COVID-19, while their clinical and radiological presentations were suggestive of PJP. Case 3 in our institution seemed to have this pattern of PJP.

In immunocompromised patients, COVID-19 is more likely to be severe [74,75,76], and concurrent IFIs should be considered in the differential diagnosis. Numerous case reports and observational studies have highlighted that fungal superinfection can complicate the clinical course of COVID-19 [77]. Although patients with IFI are expected to have underlying immunodeficiency, some data showed that patients with COVID-19 may develop IFI without preexisting immunodeficiency. The data available so far show that in concurrent IFIs, such as COVID-19 pneumonia associated pulmonary aspergillosis (CAPA), the classical IFI host risk factors may not be present [78,79]. Mortality in CAPA patients seems to be higher when compared to mortality in no-CAPA patients [21,80]. Treatment with glucocorticoids and tocilizumab for COVID-19 pneumonia may increase the risk of infection with *Aspergillus* spp. infection in severe COVID-19 patients [22].

According to the current guidelines, glucocorticoids are systematically administered to severe COVID-19 patients for 10 days or upon discharge [17,81]. Patients with a slow response to glucocorticoids may receive a long duration of treatment, which gradually enhances the risk of PJP. Other immunosuppressive drugs used to manage COVID-19 (e.g., baricitinib, tocilizumab) also increase the risk of PJP.

Notably, almost all patients in this review received systemic glucocorticoid therapy as COVID-19 treatment. One observational study by Alanio et al. [34] observed that long-term corticosteroid prescription tended to be more frequent in patients with a positive *P. jirovecii* PCR (30.0% vs. 8.2%, *p* = 0.06). This finding highlighted the role of glucocorticoid therapy in the development of PJP.

PJP is commonly associated with prolonged courses of corticosteroids (4 weeks), with a median daily dose equivalent to 30 mg of prednisone [82]. However, patients receiving doses as low as 15 mg or having baseline lymphopenia are also at risk [83]. Immunosuppression from glucocorticoids and severe COVID-19-related lymphopenia may increase the risk of SARS-CoV-2 and *P. jirovecii* concurrent infection and may occur even in individuals outside classical known risk groups. In previously colonized patients, these factors may also activate the ongoing infection with *P. jirovecii*, leading to the development of PJP.

Probably, in *P. jirovecii* superinfection, glucocorticoid therapy prescribed for COVID-19 and lymphopenia are the most important PJP risk factors (Table 2). COVID-19 patients with *P. jirovecii* airway colonization are at risk of biphasic pneumonia (i.e., superinfection), and anti-*pneumocystis* prophylaxis should be considered in case of prolonged glucocorticoid therapy or severe lymphopenia.

In addition, anti-*pneumocystis* prophylaxis should be considered during COVID-19 hospitalization of patients with classical PJP risk factors.

## 5. Conclusions

The diagnosis of PJP in patients with SARS-CoV-2 infection seems to be considerably underestimated, since the diagnosis of COVID-19 does not require BAL. Patients with concomitant PJP and COVID-19 do not necessarily have preexisting immunodeficiency. Patients with severe COVID-19 frequently develop CD4+ lymphocytopenia. They routinely require treatment with glucocorticoids and other immunosuppressive agents. Thus, severe COVID-19 appears to increase the risk of PJP. We demonstrated that detection of *P. jirovecii* in patients with COVID-19 can be characterized into three important categories (i.e., airway colonization, coinfection, and superinfection). Patients with severe COVID-19 and *P. jirovecii* colonization may require anti-*pneumocystis* prophylaxis. Observational and prospective studies are required to determine the risk of PJP in patients with moderate to severe COVID-19, considering different level of lymphopenia.

## Figures and Tables

**Table 1 jof-08-00585-t001:** Demographic and clinical characteristic of patients.

Authors and Type of Study	N of Cases	Sex, Age	Comorbidities	Immunosuppression	Radiological Findings	ACL/CD4+	Diagnostic Tests for PJP	Diagnosis EORT/MSG Cirteria	PJP Treatment, Outcome
Alanio et al. [34] Prospective observational study 2020	10	8 M, Mean age 59	DM: 3, AHT: 6, Obesity: 3, CVD:1, Asthma: 1	Long-term Steroid therapy: 3	NR	NR	PCR on RS	Probable PJP	TMP-SMX: 4 IMV: 10 Died: 3
Menon et al. [35] Case report 2020	1	F, 83	CVD, Asthma	Ulcertive Colitis, Long-term Steroid therapy	GGO Cystic Consolidation	1090/290	RT-PCR on TA BDG (305 pg/mL)	Probable PJP	TMP-SMX IMV Survived
Mang et al. [36] Case report 2020	1	M, 52	AHT, Obesity, HIV	AIDS	GGO Crazy paving	590/12	PCR on TA	Probable PJP	TMP-SMX Prednisone IMV Survived
Coleman et al. [37] Case report 2020	1	M, 54	HIV, Asthma	None	GGO Cystic	NR/422	PCR on induced sputum	Probable PJP	TMP-SMX Prednisone no IMV Survived
Bhat et al. [38] Case report 2020	1	M, 25	HIV	AIDS	GGO Cyst pneumothorax	NR/32	IFM on BAL	Probable PJP	TMP-SMX Prednisone IMV Survived
De Franceso et al. [39] Case report 2020	1	M, 65	DM, AHT CVD, SOT	SOT (kidney), Long-term steroid therapy	GGO	NR/422	PCR on sputum	Probable PJP	TMP-SMX Prednisone IMV Died
Gerber et al. [40] Case series 2020	4	F, 80	Obesity RA	RA Long-term Steroid therapy	GGO Consolidation Nodules	300/NR	PCR on BAL	Probable PJP	TMP-SMX IMV Died
		M, 70	Obesity, CKD, Hematological cancer	HSCT	GGO consolidation	222/NR	PCR on TA	Probable PJP	TMP-SMX IMV Died
		M, 83	CVD, CKD Asthma	None	GGO Crazy paving	570/NR	PCR and cytology on BAL	Proven PJP	TMP-SMX No IMV Died
		M, 80	DM, AHT CVD, Asthma	None	GGO Pleural effusion	600/NR	PCR on TA	Probable PJP	TMP-SMX IMV Died
Kronsten VT et al. [41] Case series 2021	2	M, 28	DM SOT	Liver transplant Long-term Steroid therapy	GGO	360/NR	PCR on BAL BDG on BAL 226 pg/mL	Probable PJP	TMP-SMX IMV Survived
		M, 54	SOT	Liver transplant Long-term Steroid therapy	GGO	390/NR	PCR on BAL BDG on BAL 227 pg/mL	Probable PJP	TMP-SMX IMV Survived
Gentile I et al. [42] Case series 2021	4	M, 55	None	Long-term Steroid therapy	GGO	260/62	IFM on BAL	Probable PJP	TMP-SMX No IMV Survived
		M, 75	AHT CVD	Long-term Steroid therapy	GGO	2220/1021	None	Possible PJP	TMP-SMX No IMV Survived
		F, 63	NHL	NHL Long-term Steroid therapy	GGO	240/93	IFM on BAL	Probable PJP	TMP-SMX No IMV Survived
		M, 68	None	Long-term Steroid therapy	GGO Pleural effusion	560/141	None	Possible PJP	TMP-SMX No IMV Survived
Cai et al. [43] Case report 2020	1	M, 72	RA	RA Long-term Steroid therapy	GGO Acute consolidation	340/NR	None	Possible PJP	Caspofungin No IMV Survived
Broadhurst et al. [44] Case report 2021	1	M, 54	DM AHT HIV	AIDS	GGO	690/26	IFM on sputum BDG (>500 pg/mL)	Probable PJP	TMP-SMX No IMV Died
Jeican et al. [30] Case report 2021	1	M, 52	AHT CVD Liver disease	None	Consolidation	190/NR	Positive histopathology	Proven PJP	None No IMV Died
Rubiano et al. [45] Case report 2021	1	M, 36	HIV	AIDS	GGO	400/8	PCR and IFM on BAL BDG (>500 pg/mL)	Probable PJP	TMP-SMX IMV Died
Viceconte et al. [46] Case report 2021	1	M, 54	None	Long-term Steroid therapy	GGO Consolidation	1265/895	IFM on BAL BDG positive	Probable PJP	TMP-SMX No IMV Survived
Larzábal et al. [47] Case report 2020	1	F, 46	Raynaud syndrome HIV	AIDS	Consolidation	NR/67	IFM sputum	Probable PJP	TMP-SMX No IMV Survived
Peng et al. [48] Case report 2020	1	F, 55	CVD SOT	SOT (kidney) Long-term Steroid therapy	Consolidation	310/258	PCR on sputum BDG (89.3 pg/mL)	Probable PJP	Micafungin No IMV Survived
Mouren et al. [49] Case report 2020	1	M, 65	Hematological malignancy CLL	CLL	GGO	200/NR	PCR and IFM on BAL	Probable PJP	TMP-SMX No IMV Survived
Quintana-Ortega et al. [50] Case report 2021	1	F, 11	Dermatomyositis	Dermatomyositis Long term Steroid therapy	GGO	110/NR	IFM on BAL	Probable PJP	TMP-SMX IMV Died
Anggraeni AT et al. [51] Case report 2021	1	M, 24	HIV	AIDS	GGO	176/16	none	Possible PJP	TMP-SMX No IMV Survived
Algarín-Lara H et al. [52] Case report 2021	1	M, 63	Obesity HIV	AIDS	GGO	NR/84	PCR on BAL	Probable PJP	TMP-SMX IMV Died
Merchant EA et al. [53] Case report 2021	1	M, 39	HIV	AIDS	GGO	860/1	IFM on BAL	Probable PJP	TMP-SMX IMV Died
Blaize et al. [54] Observational study 2020	2	F, 38	DM AHT	none	GGO	NR	PCR on BAL	Probable PJP	None IMV Died
		F/NR	DM AHT Obesity	none	GGO	NR	PCR on BAL	Probable PJP	None IMV Died

Abbreviations: AHT: Arterial hypertension, AIDS: acquired immunodeficiency syndrome, BDG: Beta-D-Glucan, CDK: chronic kidney disease, CVD: chronic vascular disease, DM: diabetes mellitus, GGO: ground glass opacity, HIV: Human Inmunodeficency virus, IFM: immunofluorescence microscopy, IMV: invasive mechanical ventilation, NR: not reported, RA: SOT: solid organ transplant, and TA: tracheal aspirate.

**Table 2 jof-08-00585-t002:** Patterns of COVID-19 and *P. jirovecii* colonization, coinfection, and superinfection.

Patterns	Case Classification	*P. jirovecii* Load (PCR)	Clinical Course	Radiographic Characteristics
Airway colonization	Blaize et al. [54] Alanio et al. [34]	<1000 copies/mL	Typical course of COVID-19	Similar to COVID-19 Grand-glass opacity
Superinfection	Gerber et al. [40] Gentile et al. [42] Cai et al. [43] Viceconte et al. [46] Peng et al. [48] Mouren et al. [49] Merchant EA et al. [53] Case 1 and 2 in our institution	>1000 copies/mL	Biphasic illness. The first phase is related to COVID-19, followed by clinical improvement. Subsequently, the second phase of respiratory illness (PJP) starts days or weeks later.	Similar to COVID-19 Diffuse, bilateral, interstitial infiltrates Grand-glass opacity Cysts, lobar infiltrates Solitary or multiple nodules
Coinfection	Menon et al. [35] Mang et al. [36] Coleman et al. [37] Bhat et al. [38] De Franceso et al. [39] Kronsten VT et al. [41] Broadhurst et al. [44] Jeican et al. [30] Rubiano et al. [45] Larzábal et al. [47] Anggraeni et al. [51] Cases 3 in our institution	>1000 copies/mL	Progressive disease and severe illness despite treatment for COVID-19. Admission to ICU and IMV may be required.	Similar to COVID-19 Grand-glass opacity Lobar or segmental infiltrates Cysts, nodules, effusion data

## Data Availability

Not applicable.

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
