# Peer review of "Concurrent Infection with SARS-CoV-2 and Pneumocystis jirovecii in Immunocompromised and Immunocompetent Individuals"

_jof, 2022, doi:10.3390/jof8060585_

Round 1
Reviewer 1 Report
The manuscript entitled "Pneumocystis jirovecii simultaneous infection in Covid-19 patients with or without previous immunosuppressive condition" reviewed comprehensively published literature of COVID-19 associated to PJP. This is a study that would be of interest to the medical mycology field and community. However, the manuscript needs significant improvement. The authors announce that articles concerning cases of Pneumocystis lung colonization or coinfection have been indicated. In most of evaluated cases, particularly in their institution, PCR was used for diagnosis of PJP. Indeed, the significance of detected P. jirovecii DNA using PCR alone remains uncertain and may represent colonization of the respiratory tract. The author should indicate the date of literature search for included articles and should check for updates.
Author Response
Point 1:
The authors announce that articles concerning cases of Pneumocystis lung colonization or coinfection have been indicated. In most of evaluated cases, particularly in their institution, PCR was used for diagnosis of PJP. Indeed, the significance of detected P. jirovecii DNA using PCR alone remains uncertain and may represent colonization of the respiratory tract.
Response to point 1:
We agree with the reviewer that PCR test similar to other diagnostic tests cannot 100% differentiate between colonization and disease. The differentiation between colonization and disease is made with clinical findings, imaging studies and fungal burden in qPCR. Thus, we used qPCR test results along with clinical and radiologic findings to differentiate between disease and colonization. Similar to most medical centres in North America, we use qPCR for PJP diagnosis. Data from qPCR can determine the fungal burden (DNA copies/μl) in the BAL specimens and several studies showed the fungal burden in PCR test can differentiate between colonization and disease(1). Patients with superinfection and co-infection had qPCR results >103. This level of fungal burden is in favor of disease rather than airway colonization. We included a separate section in this study and introduced airway P. jirovecii colonization in patients with COVID-19. We also reported a patient with P. jirovecii colonization.
Point 2:
The author should indicate the date of literature search for included articles and should check for updates.
Response to point 2:
We provided the information requested by the reviewer in the Materials and Methods section. We run again the literature search on April 19, 2022. We searched two databases: Pubmed and Embase. We were not able to find new case reports that we added to our article.
Please pay attention that we only included studies with a confirmed diagnosis of COVID-19 using real-time polymerase chain reaction (RT-PCR) from a nasopharyngeal or oropharyngeal swab or BAL. Conference proceedings and books were excluded.
We added an appendix and provided our search strategy (please see the attachment)
References:
- Sarasombath PT, Thongpiya J, Chulanetra M, Wijit S, Chinabut P, Ongrotchanakun J, et al. Quantitative PCR to Discriminate Between Pneumocystis Pneumonia and Colonization in HIV and Non-HIV Immunocompromised Patients. Front Microbiol [Internet]. 2021 Oct 20 [cited 2022 Apr 28];12:3090. Available from: https://www.frontiersin.org/articles/10.3389/fmicb.2021.729193/full
Reviewer 2 Report
The authors contribute a review of a rapidly evolving topic of Pneumocystis co-infection in patients with Covid-19. Further revision of newly appearing articles is recommended.
The three cases presented need to include more information like steroid doses and timing of administration, and history of Covid vaccine administration. The history of steroid use is relevant since the effects of steroids in decreasing CD4 counts are well described. The discussion on the role of steroids in inducing PcP might benefit from further discussion. This topic is referred to in the discussion section but needs some work for example in outlining the association between steroids and decreasing CD4 counts in the setting of Covid leading to PcP. Old literature shows the timing between the start of steroids and the decrease in CD4 counts. The criteria for the recommendation of anti-Pneumocystis prophylaxis are vaguely discussed and may deserve a better discussion of indications and recommendation of dose and timing in this manuscript.
The title can be shortened and focused on the purpose of the review. For example Circumstances of Pneumocystis co-infection in Covid-19.
The case definition is a value of this report and should be under a subtitle in the methods section.
minor:
Dr. Alanio's surname is misspelled on several pages.
Author Response
We divided Reviewer’s suggestions, edits, and comments into following 6 points:
Point 1:
The three cases presented need to include more information like steroid doses and timing of administration, and history of Covid vaccine administration. The history of steroid use is relevant since the effects of steroids in decreasing CD4 counts are well described
Reply to point 1:
Thanks for this important suggestion. We added that information to the revised manuscript. Please notice that we decided to change case 3 (now case 2) to superinfection group.
Point 2:
The discussion on the role of steroids in inducing PCP might benefit from further discussion. This topic is referred to in the discussion section but needs some work for example in outlining the association between steroids and decreasing CD4 counts in the setting of Covid leading to PCP. Old literature shows the timing between the start of steroids and the decrease in CD4 counts.
Reply to point 2:
Thanks for your comment. We provided a further discussion on the association between steroids and PJP in the discussion of the revised manuscript. Please see lines 312-326. PJP is commonly associated with >3 weeks of treatment with corticosteroids. In concurrent COVID-19 and infection with Pneumocystis jirovecii, immunosuppression from glucocorticosteroids and lymphopenia appear to play major roles. These two variables are the most important risk factors for PJP in patients with severe COVID-19. As the reviewer mentioned, old medical literature suggested an interval of 3-4 weeks between glucocorticoid therapy and CD4 lymphopenia; however, such an interval has not been clearly investigated. The effect of glucocorticoids on lymphocyte functions is not necessarily limited to CD4 lymphopenia. Different aspects of T cell activation, co-stimulatory pathway, IL-2 gene transcription are also affected by glucocorticoids. Hence, we did not limit our discussion to CD4 lymphopenia.
Point 3:
The criteria for the recommendation of anti-Pneumocystis prophylaxis are vaguely discussed and may deserve a better discussion of indications and recommendation of dose and timing in this manuscript
Reply to point 3:
Thanks for this important point. As discussed in the revised manuscript, PJP prophylaxis should be considered for COVID-19 patients with prolonged glucocorticoid therapy. However, lymphopenia is frequently seen in patients with COVID-19 without glucocorticoid therapy. Patients may receive other immunosuppressive drugs that associations with PJP have been shown in previous studies (e.g., baricitinib). On the other hand, many patients in medical literature developed PJP with a short duration of glucocorticoid therapy. Therefore, the level of evidence is not satisfactory for determining the risk threshold and, subsequently, PJP prophylaxis. Observational studies are required to determine the risk of PJP in patients with COVID-19 and consequently establish prophylaxis criteria.
Point 4:
The title can be shortened and focused on the purpose of the review. For example Circumstances of Pneumocystis co-infection in Covid-19.
Reply to point 4:
Thanks for the suggestion. We modified the title to “Concurrent Infection with SARS-CoV-2 And Pneumocystis jirovecii in immunocompromised and immunocompetent individuals”.
Point 5:
The case definition is a value of this report and should be under a subtitle in the methods section.
Rely to point 5:
Thanks for the suggestion, we added a new paragraph to the method section (lines121-129).
Point 6:
Dr. Alanio's surname is misspelled on several pages.
Reply to point 6:
Thanks, we corrected it.
Reviewer 3 Report
The manuscript from Gioia, et al., is a review of cases of Pneumocystis jirovecii pneumonia (PjP) in COVID-19 patients with an added description of 3 more cases at the author’s institution. The data is well researched and presented. The authors conclude that there are three scenarios in which P. jirovecii is found in COVID-19 patients with the most invasive being in patients who are immunosuppressed with glucocorticoid treatment. The references look reasonable and manuscripts were searched in three languages. The manuscript would be readable for a wider audience if the authors provided a more complete list of acronyms/abbreviations. For a non-clinician researcher, some of the clinical acronyms used in the manuscript and tables are not obvious. Examples include GGO, IMV, and AHT. These could be inserted in the abbreviations list at the end of Table 1 and defined upon first use in the manuscript. Since Table 1 is multiple pages long, it would be more readable if the column titles were repeated for each page. Overall, the manuscript provides information that both clinicians and researchers should know regarding the potential for PjP in COVID-19 patients.
Author Response
We divided these important comments into 2 points as follows:
Point 1:
The manuscript would be readable for a wider audience if the authors provided a more complete list of acronyms/abbreviations. For a non-clinician researcher, some of the clinical acronyms used in the manuscript and tables are not obvious. Examples include GGO, IMV, and AHT. These could be inserted in the abbreviations list at the end of Table 1 and defined upon first use in the manuscript.
Reply to point 1:
Considering Journal of Fungi Instruction to Author guidance, we defined the acronyms the first time they appeared in the text. We also provided a list of acronyms at the end of Table 1.
Point 2:
Since Table 1 is multiple pages long, it would be more readable if the column titles were repeated for each page
Reply to pint 2:
We corrected the table as recommended.
Round 2
Reviewer 1 Report
The manuscript improved after attention and correct the comments.
Author Response
Thanks
Reviewer 2 Report
The manuscript improved and I consider is adequate for publication.
Author Response
Thanks